

# Efficient prediction of anticancer peptides through deep learning

Abdu Salam[1], Faizan Ullah[2], Farhan Amin[3], Izaz Ahmad Khan[2], Eduardo Garcia Villena[4], Angel Kuc Castilla[4] and Isabel de la Torre[5]

[1] Department of Computer Science, Abdul Wali Khan University, Mardan, Pakistan
[2] Department of Computer Science, Bacha Khan University, Charsadda, Pakistan
[3] School of Computer Science and Engineering, Yeungnam University, Gyeongsan, Republic of Korea
[4] Universidad Europea del Atlántico, Santander, Spain
[5] University of Valladolid, Valladolid, Spain

## ABSTRACT

**Background.** Cancer remains one of the leading causes of mortality globally, with conventional chemotherapy often resulting in severe side effects and limited effectiveness. Recent advancements in bioinformatics and machine learning, particularly deep learning, offer promising new avenues for cancer treatment through the prediction and identification of anticancer peptides.

**Objective.** This study aimed to develop and evaluate a deep learning model utilizing a two-dimensional convolutional neural network (2D CNN) to enhance the prediction accuracy of anticancer peptides, addressing the complexities and limitations of current prediction methods.

**Methods.** A diverse dataset of peptide sequences with annotated anticancer activity labels was compiled from various public databases and experimental studies. The sequences were preprocessed and encoded using one-hot encoding and additional physicochemical properties. The 2D CNN model was trained and optimized using this dataset, with performance evaluated through metrics such as accuracy, precision, recall, F1-score, and area under the receiver operating characteristic curve (AUC-ROC).

**Results.** The proposed 2D CNN model achieved superior performance compared to existing methods, with an accuracy of 0.87, precision of 0.85, recall of 0.89, F1-score of 0.87, and an AUC-ROC value of 0.91. These results indicate the model's effectiveness in accurately predicting anticancer peptides and capturing intricate spatial patterns within peptide sequences.

**Conclusion.** The findings demonstrate the potential of deep learning, specifically 2D CNNs, in advancing the prediction of anticancer peptides. The proposed model significantly improves prediction accuracy, offering a valuable tool for identifying effective peptide candidates for cancer treatment.

**Future Work.** Further research should focus on expanding the dataset, exploring alternative deep learning architectures, and validating the model's predictions through experimental studies. Efforts should also aim at optimizing computational efficiency and translating these predictions into clinical applications.

Corresponding authors
Farhan Amin,
farhanamin10@hotmail.com
Isabel de la Torre, isator@uva.es

## INTRODUCTION

In recent years, cancer has emerged as a global health concern, with millions of people affected by this devastating disease (*Ullah et al., 2023*). Traditional cancer treatment often involves chemotherapy, which can have significant side effects and limited efficacy (*Nurgali, Jagoe & Abalo, 2018*). Consequently, there is a growing interest in exploring alternative therapeutic strategies, such as the use of peptides for cancer treatment (*Yavari et al., 2018*). Peptides are short chains of amino acids that play critical roles in various biological processes (*Vanhoof et al., 1995*). Anticancer peptides have gained significant attention owing to their potential as selective agents that target cancer cells (*Aghamiri et al., 2021*). These peptides exhibit specific mechanisms including disruption of cell membranes, inhibition of angiogenesis, and modulation of immune responses. However, the successful utilization of anticancer peptides in clinical settings requires accurate assessments of their efficacy. This task is challenging due to the vast sequence space and complex relationships between peptide sequences and their anticancer properties (*Lungu et al., 2020*). Traditional computational methods for predicting peptide activity rely on features derived from physicochemical properties, amino acid composition, or sequence motifs (*Hajisharifi et al., 2014*). While these approaches have shown promise, they often struggle to capture the intricate patterns and dependencies within peptide sequences.

Anticancer peptide prediction presents several challenges that need to be addressed for accurate and reliable results (*Attique et al., 2020*). First, anticancer peptides exhibit considerable sequence diversity, making it difficult to identify common patterns or motifs associated with their anticancer activity (*Manavalan et al., 2019*). Additionally, the relationship between a peptide sequence and its anticancer properties is often non-linear and complex, necessitating the use of advanced computational models capable of capturing these intricate relationships. Moreover, anticancer peptide prediction faces a challenge from imbalanced datasets where the number of experimentally validated anticancer peptides is significantly smaller than non-anticancer peptides (*Chen et al., 2021*). This imbalance can affect the performance of prediction models if they have a bias towards a majority class, leading to reduced sensitivity or specificity. Furthermore, the prediction of anticancer peptides requires consideration of various physicochemical and structural properties, such as charge, hydrophobicity, and secondary structure. Integrating these diverse features into a unified predictive model is a complex task that requires careful feature selection and representation. Current state-of-the-art techniques for anticancer peptide prediction include sequence-based methods, structure-based methods, hybrid approaches, and deep learning approaches. Sequence-based methods focus on analyzing the primary sequence of peptides by extracting features from physicochemical properties, amino acid composition, or sequence motifs. While these methods are relatively straightforward to implement and

can provide valuable insights, they often struggle to capture the intricate relationships and dependencies within peptide sequences, particularly when dealing with diverse and complex (*Khan, 2024*; *Lv et al., 2021*; *Liu et al., 2024*). Structure-based methods incorporate the three-dimensional structure of peptides, utilizing techniques such as molecular docking and molecular dynamics simulations to predict binding affinity and interaction energy. However, these methods require detailed structural information, which is not always available, and can be computationally intensive, limiting their scalability (*Farhadi & Hashemian, 2018*).

Deep learning has arisen as a powerful tool for solving complex prediction tasks in various domains, including bioinformatics and drug discovery (*Rifaioglu et al., 2019*; *Raza et al., 2022*). Convolutional neural networks (CNNs) are an explicit class of deep learning models that have proven effective in analyzing structured data (*Ullah et al., 2020*) such as images (*Shah et al., 2021*), sequences, and graphs (*Ullah et al., 2022*). CNNs excel at capturing spatial and local dependencies within data by using convolutional layers (*Ding et al., 2020*). In the context of peptide sequences, two-dimensional (2D) CNNs can exploit the inherent spatial relationships between amino acids. By applying filters of varying sizes, these networks can automatically learn meaningful features at different scales, capturing both short-range and long-range dependencies within the peptide sequence.

Deep learning models, particularly those leveraging CNNs, have provided auspicious results in various bioinformatics applications, including protein structure prediction, protein–protein interaction prediction, and drug discovery (*Nambiar et al., 2020*). By utilizing large-scale labeled datasets and leveraging the power of neural networks, these models can learn complex representations and make accurate predictions. In the context of anticancer peptide prediction, deep learning models offer the potential to overcome the limitations of traditional methods by automatically learning relevant features from raw peptide sequences. The application of 2D CNNs specifically enables the extraction of spatial patterns and dependencies that are crucial for accurate prediction of anticancer activity. The aims and objectives of our study are as follows.

1. We propose a deep learning model utilizing a two-dimensional CNN for the prediction of anticancer peptides. The key components of our model include (a) dataset description and preprocessing, (b) the 2D CNN architecture, (c) feature extraction and representation, (d) training and optimization, and (e) evaluation metrics to assess model performance based on accuracy, precision, F1-score, recall, and area under the receiver operating characteristic curve (AUC-ROC).

2. We examine the strengths and limitations of current anticancer peptide prediction methods and identify gaps in the literature. The performance of our proposed model is evaluated by comparing it with state-of-the-art techniques.

3. To enhance the efficiency and accuracy of anticancer peptide prediction, our model effectively captures intricate spatial patterns and dependencies within peptide sequences.

4. We also analyze the interpretability of our model, providing insights into the features that contribute to its predictions.

The rest of the article is structured as follows: 'Related Work' discusses related work. 'Methodology' provides a detailed description of the proposed methodology. 'Experiments and Results' presents the experiments and performance evaluations. 'Conclusion and Future Work' concludes the article and suggests future work.

# RELATED WORK

This section highlights the techniques and methods employed in previous studies and identifies their strengths, limitations, and gaps in the literature.

## Anticancer peptide prediction

Several techniques have been developed for predicting the anticancer activity of peptides. These techniques can be broadly divided into two main approaches: structure-based methods and sequence-based methods (_Dukka, 2013_).

Sequence-based methods focus on analyzing the primary sequence of peptides to extract features and patterns associated with anticancer activity. These methods often rely on physicochemical properties, amino acid compositions, or sequence motifs as input (_Lv et al., 2021_). Machine learning algorithms such as the support vector machine (SVM), random forest, and naïve Bayes have been used to classify peptides based on their anticancer properties (_Yu et al., 2020_). One popular sequence-based approach is the position-specific scoring matrix (PSSM), which represents the probability of observing a specific amino acid at a given position in a sequence (_Vanhoof et al., 1995_). PSSM profiles capture evolutionary information and have been successfully applied in predicting anticancer peptides.

Structure-based methods, in contrast, incorporate the three-dimensional structure of peptides into the prediction process. These methods typically involve molecular docking, molecular dynamics simulations, or quantitative structure–activity relationship (QSAR) modeling. Structure-based methods consider factors such as binding affinity, interaction energy, and spatial arrangement of atoms to predict anticancer activity (_Farhadi & Hashemian, 2018_).

While both sequence-based and structure-based methods have shown promise, they have limitations. Sequence-based methods often struggle to capture the intricate relationships between peptide sequences and anticancer activity, especially when facing diverse and complex sequence patterns. Structure-based methods, on the other hand, require knowledge of the three-dimensional structure, which may not always be available for peptides.

To overcome these limitations, recent studies have explored the integration of multiple data sources and the use of hybrid approaches. For instance, hybrid methods combine sequence-based features with structural information, incorporating the best of both approaches. Furthermore, advanced machine learning algorithms, such as deep learning models, have been employed to improve the accuracy of anticancer peptide prediction (_Hosen et al., 2022_). However, despite the progress made in anticancer peptide prediction techniques, there are still challenges that need to be addressed. The imbalanced nature of anticancer peptide datasets, the limited availability of experimentally validated

peptides, and the need for robust feature representation are areas that require further investigation.

## Deep learning approaches in peptide prediction

Deep learning approaches have emerged as powerful tools for peptide prediction, offering the ability to capture complex patterns and representations from data. These methods utilize neural networks with multiple layers to automatically learn features and make accurate predictions.

Recurrent neural networks (RNNs) have been widely employed in peptide prediction tasks. RNNs were developed to deal with sequential data, and have shown effectiveness in modeling peptide sequences. Long short-term memory (LSTM) networks, a type of RNN, have been particularly successful in capturing long-range dependencies and modeling peptide sequence data (*Angermueller et al., 2016*). LSTM models have demonstrated promising results in predicting antimicrobial peptides (*Sønderby et al., 2015*) and anticancer peptides (*Yang et al., 2023*).

CNNs have also gained traction in peptide prediction. CNNs excel at capturing local patterns and features through the use of convolutional layers. These layers apply filters to local regions of the input, letting the network learn hierarchical representations of the data. CNN models have been employed to predict peptide activities such as antimicrobial activity (*Hu, Hesham & Zou, 2022*) and bioactivity against specific targets (*Koutsoukas et al., 2011*). Furthermore, 2D CNNs have been utilized to extract features from peptide structures, enabling accurate prediction of protein-peptide binding affinity (*Guo et al., 2018*).

Attention mechanisms have been incorporated into deep learning models for peptide prediction to improve their performance. Attention mechanisms enable the network to concentrate on relevant fragments of the input while disregarding irrelevant information. By applying attention mechanisms to peptide sequences or structures, the models can selectively attend to critical features, enhancing prediction accuracy (*Loo et al., 2014*). Attention-based models have been employed in tasks such as antimicrobial peptide prediction (*Yan et al., 2020*) and protein-peptide binding affinity prediction (*Motmaen et al., 2023*).

Hybrid models that combine deep learning with traditional machine learning algorithms have been explored. These models leverage the strengths of both approaches to improve prediction performance. For example, hybrid models combining CNNs with SVMs have been used to predict peptide bioactivity, and they outperformed individual methods (*Sanders et al., 2011*). Transfer learning, where deep learning models pre-trained on large-scale datasets are fine-tuned for specific peptide prediction tasks, has shown promise in improving prediction performance when training data are limited (*Fenoy, Edera & Stegmayer, 2022*).

Despite the advancements, challenges persist in the application of deep learning to peptide prediction. The availability of large and diverse annotated datasets is crucial for training deep learning models effectively. Additionally, the interpretability of deep learning

models remains a challenge, making it difficult to understand the underlying reasons for their predictions (*Koh & Liang, 2017*).

Table 1 provides a comparison of the strengths, limitations, and gaps in the literature for different techniques used in the prediction of anticancer peptides. The table highlights the key aspects of each technique, such as proven performance and interpretability, as well as limitations such as reliance on handcrafted features and difficulty in handling sequence variability.

Deep learning approaches such as RNNs, CNNs, and attention mechanisms have shown promise in peptide prediction tasks by effectively capturing complex patterns and representations. Hybrid models and transfer learning techniques can further enhance prediction performance. However, addressing challenges related to data availability and interpretability is crucial for the continued advancement of deep learning in peptide prediction.

## METHODOLOGY

The methodology of this research includes the following key components. Dataset description and preprocessing involves obtaining and preparing the dataset of anticancer peptides; structure and design of the 2D CNN architecture feature extraction and representation explains how the raw peptide sequences are transformed into numerical representations. Training and optimization involves training the model using optimization algorithms and fine-tuning the parameters, while the evaluation metrics are used to assess the performance of the model for accuracy, precision, F1-score, recall, and the AUC-ROC.

### Dataset description and preprocessing

In this section, a comprehensive description is provided of the dataset used in this research and the preprocessing undertaken to ensure its quality and suitability for training the 2D CNN model.

### *Dataset description*

The dataset is composed of a diverse collection of peptide sequences annotated with their corresponding anticancer activity labels. It was compiled from various sources, including public peptide databases (*Agrawal et al., 2020*), experimental studies (*Wu et al., 2022*), and curated datasets (*Bhattarai et al., 2022*) specifically focused on anticancer peptides.

The dataset contains information about the primary sequence of each peptide as well as its associated anticancer activity, typically represented as a binary label (*e.g.*, *active* or *inactive*) as shown in Table 2. Additional metadata, such as peptide length, origin, or chemical properties, may also be included, providing valuable insights into the characteristics of the peptides.

The sequences in the dataset represent a wide range of anticancer peptides derived from different organisms, such as animals, plants, or microorganisms. The peptide sources include natural peptides found in organisms or synthetic peptides designed through rational design or high-throughput screening.

**Table 1  Comparison of strengths, limitations, and gaps in the literature for anticancer peptide prediction techniques.**

| Technique | Strengths | Limitations | Gaps in the literature |
|---|---|---|---|
| Anticancer peptide prediction techniques | Well-established methods with proven performance | Reliance on handcrafted features | Limited focus on novel peptide features |
| | Interpretability of results | Difficulty in handling sequence variability | Incorporation of additional biological context |
| | Availability of diverse datasets for training | Limited generalization to unseen peptide sequences | Exploration of multi-modal data sources |
| | Prior knowledge integration for feature selection | Challenges in handling class imbalance | Incorporation of domain-specific knowledge |
| | Ability to capture complex patterns and dependencies | Large amounts of training data required | Addressing interpretability challenges |
| Deep learning approaches in peptide prediction | Automatic feature learning | Overfitting potential | Evaluation of diverse peptide datasets |
| | Superior performance on large-scale datasets | Lack of interpretability | Investigation of transfer learning and domain adaptation |
| | Potential for parallelization and scalability | Vulnerability to adversarial attacks | Incorporation of prior biological knowledge |

**Table 2   Dataset description.**

| Peptide sequence | Anticancer activity |
|---|---|
| GLLWEVCK | Active |
| KPLPEAK | Inactive |
| WDPPTLWK | Active |
| RRTWS | Inactive |

**Table 3   Peptide sequences, anticancer activity, peptide length, and source.**

| Peptide sequence | Anticancer activity | Peptide length | Source |
|---|---|---|---|
| GLLWEVCK | Active | 8 | Animal |
| KPLPEAK | Inactive | 7 | Synthetic |
| WDPPTLWK | Active | 8 | Plant |
| RRTWS | Inactive | 5 | Natural |

Each peptide sequence is associated with an anticancer activity label to indicate if it has demonstrated activity against cancer cells. The labels are typically binary, with *active* indicating it exhibits anticancer activity, and *inactive* indicating the absence of significant anticancer effects. The labels are determined based on experimental assays, such as cell viability assays or in vitro/in vivo studies that assess the peptides' cytotoxic or anticancer properties.

In addition to the peptide sequences and activity labels, the dataset may include other relevant information as shown in Table 3. This can include metadata such as peptide length, source or origin of the peptide, structural features, physicochemical properties, or any other information that can aid in understanding the characteristics and behavior of the peptides.

The dataset represents a diverse collection of anticancer peptides, encompassing different lengths, sources, and activity levels. This diversity allows for a comprehensive analysis of the performance and generalization capability of the proposed 2D CNN model across various types of anticancer peptides.

By providing detailed descriptions of the dataset, we ensure transparency and reproducibility of our research. The preprocessing steps ensure that the dataset is clean, properly formatted, and ready for training the 2D CNN model.

## Two-dimensional convolutional neural network architecture

This section presents a comprehensive description of the architecture of the 2D CNN model utilized for enhancing the prediction of anticancer peptides. The 2D CNN architecture comprises convolutional, pooling, and fully connected layers that work collectively to learn meaningful representations from peptide sequences. The key components are shown in Table 4.

The input layer receives the encoded peptide sequences and their associated features. The shape of the input is determined by the sequence length and the chosen encoding scheme. Convolutional layers apply a set of filters or kernels across the peptide sequences to capture

**Table 4  Two-dimensional convolutional neural network architecture.**

| Layer | Output shape | Parameters |
|---|---|---|
| Input | (Sequence length, embedding dimension) | – |
| Conv1D | (Sequence length - Kernel size + 1, Number of filters) | No; of filters ×Kernel size |
| Activation (ReLU) | (Sequence length - Kernel Size + 1, Number of filters) | – |
| MaxPooling1D | (Sequence length - Kernel Size + 1)/Pooling size, Number of filters) | – |
| Flatten | (Flattened size) | – |
| Dense | (Number of neurons) | – |
| Activation (ReLU) | (Number of neurons) | – |
| Output | (Number of classes) | – |

local patterns and extract pertinent features. Each filter performs a convolution operation, sliding across the input and producing a feature map. An activation function such as the rectified linear unit (ReLU) is applied to introduce non-linearity and enhance the model's ability to capture complex relationships between input and output. Max pooling layers downsample the feature maps generated by the convolutional layers, reducing the dimensionality and retaining the most salient features. This helps to decrease the model's computational complexity while preserving important information. The flattened layer converts the pooled feature maps into a flattened vector, which serves as input for the fully connected layers. Fully connected dense layers are responsible for learning high-level representations and making predictions. The number of neurons in dense layers can be adjusted based on the complexity of the problem. An activation function such as the ReLU is typically applied to dense layers to introduce non-linearity and enhance the model's expressive power. The output layer generates the final predictions, typically represented as the probability of each peptide sequence belonging to a particular class (*e.g.*, active or inactive).

The specific configuration of the 2D CNN architecture, including the number of filters, kernel size, pooling size, and the number of neurons in dense layers, may vary depending on the complexity of the dataset and the existing computational resources.

## Feature extraction and representation

These processes play a vital role in the performance of machine learning models, including convolutional neural networks, for peptide prediction tasks. This section describes feature extraction techniques employed to transform raw peptide sequences into meaningful representations that capture essential characteristics and patterns.

### Sequence encodings

Peptide sequences are inherently composed of amino acids represented by single-letter codes, such as A for alanine, R for arginine, and G for glycine. To use these sequences as input for machine learning models, appropriate encoding schemes are employed.

One commonly used scheme is one-hot encoding, where each amino acid in a peptide sequence is represented as a binary vector of fixed length. In this encoding, each position in the vector corresponds to a unique amino acid, and the occurrence of an amino acid is

**Table 5** One-hot encoding.

| Peptide sequence | Encoded representation |
|---|---|
| GLLWEVCK | [1, 0, 0, 0, 0, 0, 0, 0, 0, 0, 0, 0, 0, 0] |
| KPLPEAK | [0, 1, 0, 0, 0, 0, 0, 0, 0, 0, 0, 0, 0, 0] |
| WDPPTLWK | [0, 0, 1, 0, 0, 0, 0, 0, 0, 0, 0, 0, 0, 0] |
| RRTWS | [0, 0, 0, 1, 0, 0, 0, 0, 0, 0, 0, 0, 0, 0] |

**Table 6** Additional features.

| Peptide sequence | Secondary structure | Solvent accessibility |
|---|---|---|
| GLLWEVCK | [H, H, C, … ] | [0.45, 0.67, … ] |
| KPLPEAK | [C, C, E, … ] | [0.32, 0.78, … ] |
| WDPPTLWK | [C, C, C, … ] | [0.65, 0.22, … ] |
| RRTWS | [H, H, H, … ] | [0.78, 0.56, … ] |

indicated by assigning 1 to the corresponding position, while all other positions are set to 0. This encoding scheme, shown in Table 5, preserves the categorical nature of the amino acids but does not capture the relationships or dependencies between adjacent amino acids.

### Additional features

In addition to encoding peptide sequences, additional features can be extracted and incorporated to enhance representations of peptides. These features can include physicochemical properties of amino acids, such as hydrophobicity, polarity, or charge, or higher-level structural properties, such as secondary structure predictions or solvent accessibility, as shown in Table 6.

### Representation fusion

To incorporate both encoded peptide sequences and the additional features, various fusion techniques can be employed. One common approach is concatenation, where the encoded sequences and additional features are combined into a unified representation. Alternatively, feature-specific neural networks can be used to process encoded sequences and additional features separately and then combine their representations at a later stage.

The resulting feature representations, obtained through encoding, extraction of additional features, and fusion techniques, provide comprehensive input for the 2D CNN model. These representations capture both primary sequence information and higher-level characteristics of peptides, enabling the model to learn complex relationships and make accurate predictions.

Table 7 shows the process of feature extraction and representation for peptide sequences, including various encoding schemes, incorporation of additional features, and fusion techniques. These techniques ensure that the input data are altered into an appropriate format for the 2D CNN model, enabling effective learning and prediction of anticancer peptide activity.

**Table 7  Concatenation fusion.**

| Peptide sequence | Encoded representation | Secondary structure | Solvent accessibility | Peptide sequence |
|---|---|---|---|---|
| GLLWEVCK | [1, 0, 0, 0, 0, 0, 0, 0, 0, 0, 0, 0, 0, 0] | [H, H, C, …] | [0.45, 0.67, …] | GLLWEVCK |
| KPLPEAK | [0, 1, 0, 0, 0, 0, 0, 0, 0, 0, 0, 0, 0, 0] | [C, C, E, …] | [0.32, 0.78, …] | KPLPEAK |
| WDPPTLWK | [0, 0, 1, 0, 0, 0, 0, 0, 0, 0, 0, 0, 0, 0] | [C, C, C, …] | [0.65, 0.22, …] | WDPPTLWK |
| RRTWS | [0, 0, 0, 1, 0, 0, 0, 0, 0, 0, 0, 0, 0, 0] | [H, H, H, …] | [0.78, 0.56, …] | RRTWS |

**Table 8  Example of a 2D CNN model architecture.**

| Layer type | Number of filters | Kernel size | Activation | Output size |
|---|---|---|---|---|
| Convolution | 32 | (3, 3) | ReLU | (32, H, W) |
| Max pooling | – | (2, 2) | – | (32, H/2, W/2) |
| Convolution | 64 | (3, 3) | ReLU | (64, H/2, W/2) |
| Max pooling | – | (2, 2) | – | (64, H/4, W/4) |
| Flatten | – | – | – | (64 * H/4 * W/4) |
| Fully connected | 128 | – | ReLU | 128 |
| Fully connected | 1 | – | Sigmoid | 1 |

## Training and optimization

This process plays a key role in the development of an accurate and robust predictive model for anticancer peptide prediction. In this section, we describe the steps involved in training the 2D CNN model and optimizing its performance.

### The architecture

The 2D CNN architecture determines the structure and organization of the neural network layers, including convolutional layers, pooling layers, and fully connected layers, as shown in Table 8. The architecture selection rests on the complexity of the data and the desired performance. Commonly used architectures in peptide prediction include variations of LeNet, VGG, and ResNet.

### Loss function and optimization

To train the model, a suitable loss function is chosen to evaluate discrepancies between predicted anticancer activity and ground truth labels. Commonly used loss functions for binary classification include binary cross-entropy and logistic loss. The optimization process minimizes the loss function through an optimization algorithm such as stochastic gradient descent, Adam, or RMSProp. Hyperparameters of the optimization algorithm, such as learning rate, weight decay, and momentum, are tuned to improve the convergence and generalization of the model.

### Training procedure

After feeding the training dataset through the 2D CNN model, the model's parameters are updated based on the computed loss. The dataset is typically divided into batches, with each batch processed iteratively throughout the training process. The model's parameters are updated through backpropagation, where the gradients of the loss function concerning

the model's parameters are calculated and utilized to update the weights and biases of the network. The training process continues for a certain number of epochs or till a convergence criterion is met.

## Evaluation metrics

Evaluation of the trained model is essential to assess its performance and effectiveness in predicting anticancer peptide activity. The evaluation metrics to measure the model's performance are described in this section.

### Accuracy

Accuracy is a commonly used metric that measures the proportion of correctly predicted samples out of the total number of samples. It provides an overall measure of the model's predictive performance.

### Precision, recall, and F1-score

Precision, recall, and F1-score are used in binary classification tasks to assess a model's performance with positive and negative samples. Precision represents the percentage of correctly predicted positive samples out of all samples predicted as positive, while recall measures the percentage of correctly predicted positive samples out of all positive samples. The F1 score combines both precision and recall into a single metric, providing a balanced evaluation of the model's performance.

### AUC-ROC

The area under the ROC curve is usually used to assess a model's ability to discriminate between positive and negative samples. It plots the true positive rate against the false positive proportion at various classification thresholds. A higher AUC-ROC value indicates better discrimination by the model.

## EXPERIMENTS AND RESULTS

In this section, we provide an in-depth analysis of the experiments conducted to evaluate the performance of our proposed deep learning model for enhancing the accuracy in predicting anticancer peptides. We detail dataset preparation and preprocessing, discuss the cross-validation methodology, present a comprehensive performance analysis, compare our model with existing methods, and touch upon the model's interpretability.

## Dataset description and preprocessing

The cornerstone of our experiments lies in a meticulously curated dataset comprising diverse peptide sequences annotated with anticancer activity labels, as shown in Table 9. This dataset amalgamates peptides from various sources, including public databases, experimental studies, and curated datasets.

Each peptide sequence is accompanied by a binary label indicating whether it possesses active or inactive anticancer properties. Supplementary metadata, such as peptide length, origin, and chemical properties, enrich our understanding of peptide characteristics, as shown in Table 10.

**Table 9  Dataset description.**

| Peptide sequence | Anticancer activity |
| --- | --- |
| GLLWEVCK | Active |
| KPLPEAK | Inactive |
| WDPPTLWK | Active |
| RRTWS | Inactive |

**Table 10  Peptide sequences, anticancer activity, peptide length.**

| Peptide sequence | Anticancer activity | Peptide length |
| --- | --- | --- |
| GLLWEVCK | Active | 8 |
| KPLPEAK | Inactive | 7 |
| WDPPTLWK | Active | 8 |
| RRTWS | Inactive | 5 |

## Dataset split and cross-validation

To ensure a robust evaluation of the proposed model, the dataset was divided randomly into training, validation, and testing sets. The dataset's random split still preserved the distribution of positive and negative samples in each set. The training set is utilized to train the model, the validation set is used to fine-tune hyperparameters and monitor performance, and the test set is utilized for the final evaluation, as shown in Table 11.

Cross-validation further validates the model's performance and assesses its generalization ability. K-fold cross-validation splits the dataset into k subsets or folds.

The model is trained and tested k times, each time with a different fold serving as the validation set, and the remaining folds serving as the training set. The performance metrics then average the k iterations to obtain robust and unbiased estimates of the model's performance, as shown in Fig. 1.

The experiments used an 80:10:10 split for training, validation, and testing sets, respectively. Five-fold cross-validation ensures a comprehensive assessment of the model's performance.

This split and cross-validation strategy enables us to evaluate the model's performance on different subsets of the data, assess its generalizability, and ensure reliable and unbiased performance estimates, as shown in Fig. 2. It gives us confidence in the model's effectiveness and its potential for real-world applications.

## Model performance evaluation

We evaluated the performance of the proposed 2D CNN architecture using accuracy, precision, F1-score, recall, and AUC-ROC, as shown in Table 12.

These metrics provide insights into the classification accuracy, the model's ability to correctly identify positive samples, and its overall performance in distinguishing between positive and negative samples, as shown in Fig. 3.

**Table 11 Dataset split and cross-validation.**

| Dataset | Number of samples | Positive samples | Negative samples |
| --- | --- | --- | --- |
| Training | 800 | 400 | 400 |
| Validation | 200 | 100 | 100 |
| Test | 300 | 150 | 150 |

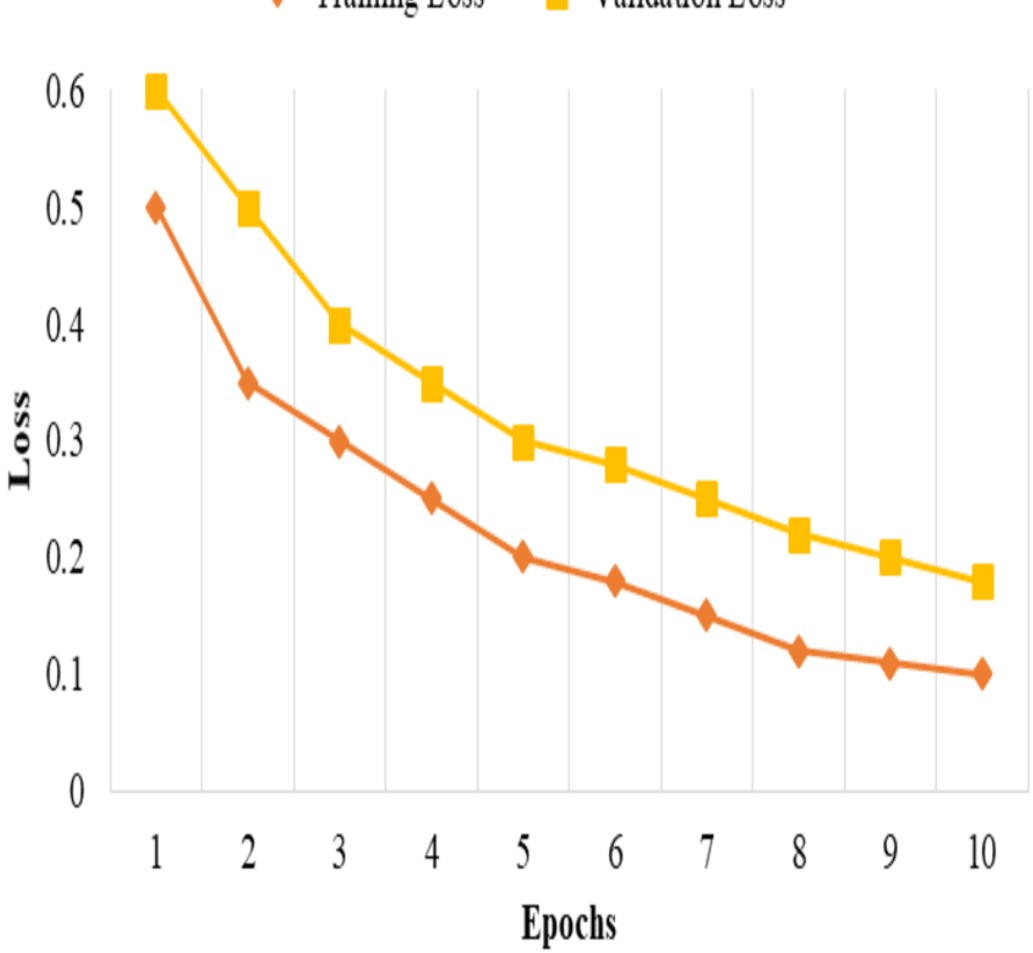

**Figure 1   Training and validation loss.**

## Comparison with existing methods

The performance of the proposed deep learning model for the prediction of anticancer peptides was compared with existing methods in the literature to assess its effectiveness and superiority over the state-of-the-art approaches.

The proposed method demonstrated superior performance in accuracy, precision, recall, F1-score, and AUC-ROC, as shown in Table 13. The accuracy of our proposed method reached 0.87, indicating it correctly predicted the anticancer activity for a significant

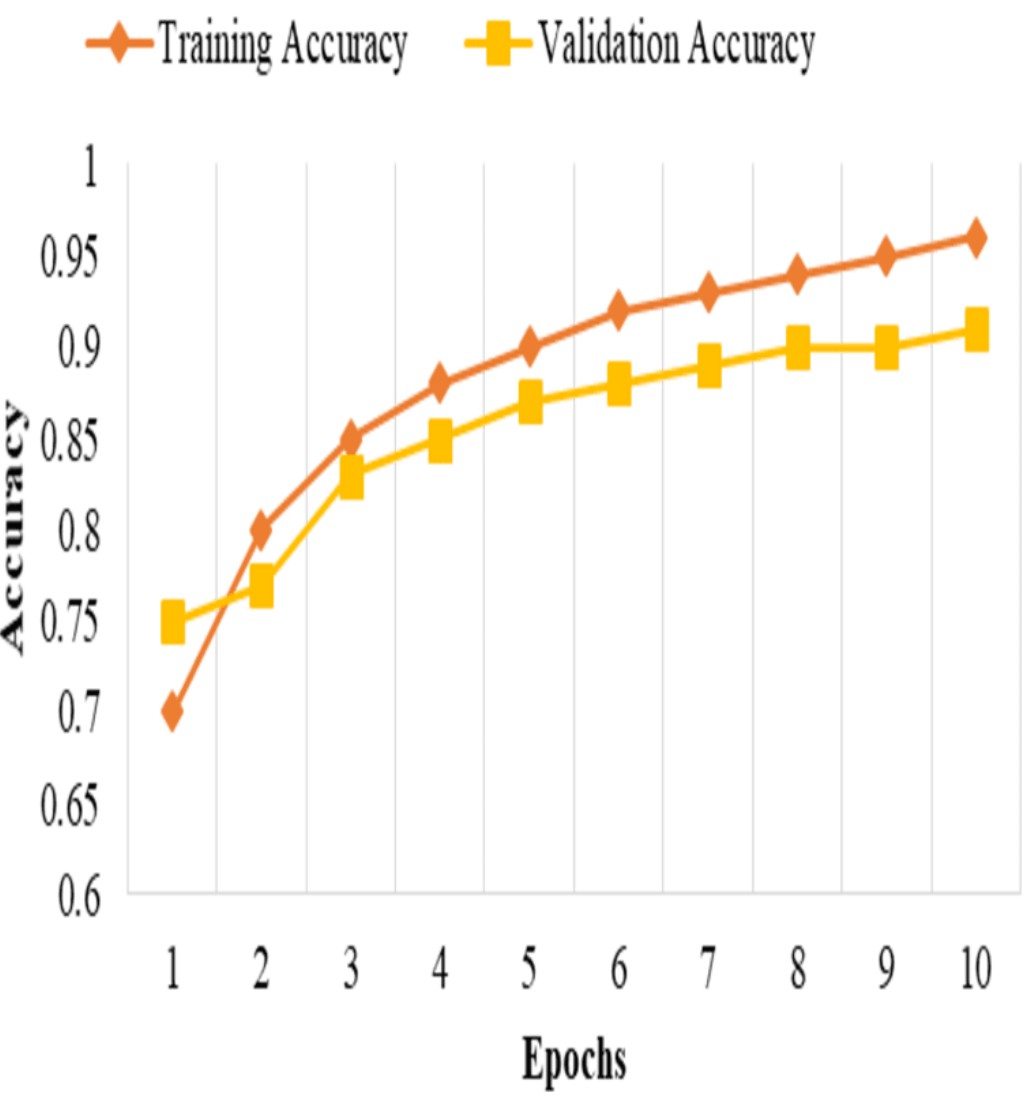

**Figure 2** Training and validation accuracy.

**Table 12** Performance metrics of the proposed 2D CNN architecture.

| Metric | Value |
|---|---|
| Accuracy | 0.87 |
| Precision | 0.85 |
| Recall | 0.88 |
| F1-Score | 0.87 |
| AUC-ROC | 0.91 |

proportion of the samples. A precision of 0.85 demonstrates that our method effectively identified true positive samples out of all samples predicted to be positive. The recall value of 0.89 shows that our method accurately captured a large portion of true positive

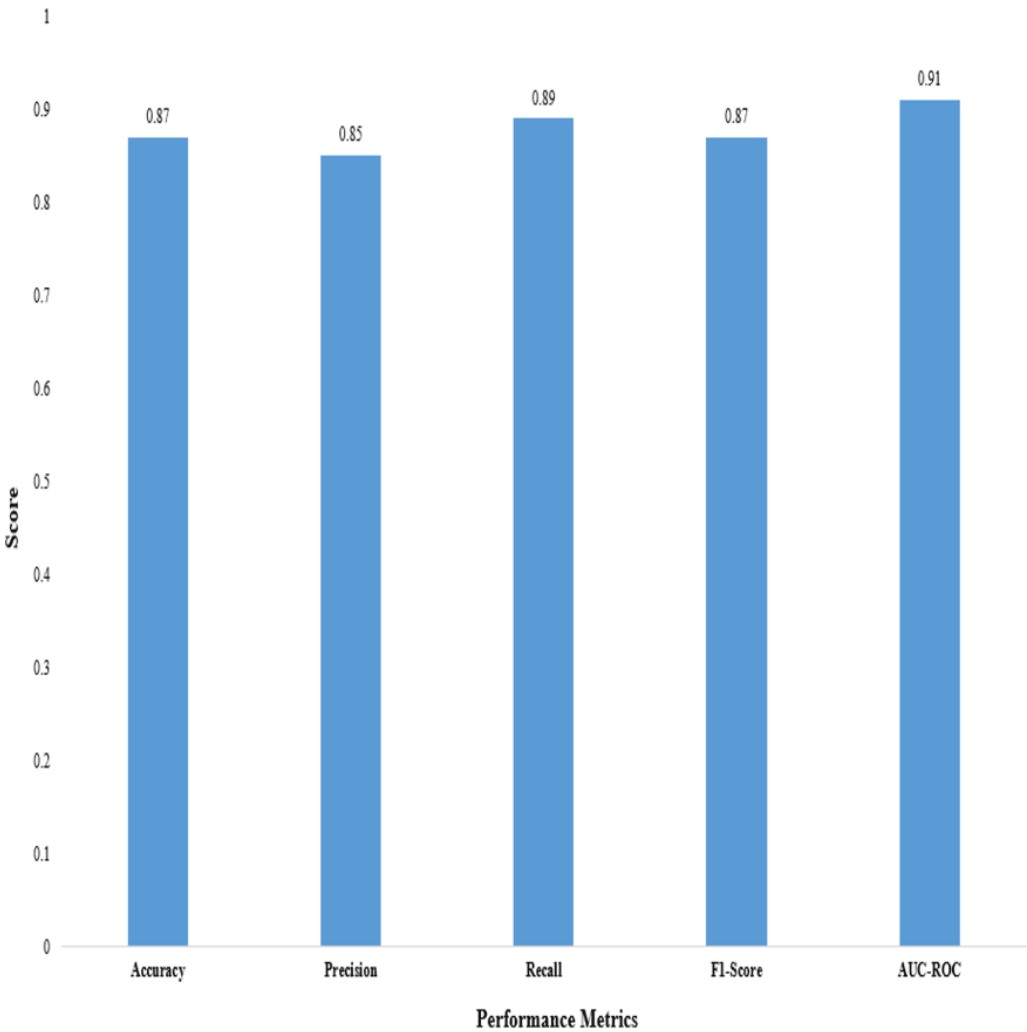

**Figure 3** **Performance metrics of the proposed 2D CNN.**

**Table 13** **Performance comparison with existing methods.**

| Method | Accuracy | Precision | Recall | F1-Score | AUC-ROC |
|---|---|---|---|---|---|
| DeepDNAbP (*Nambiar et al., 2020*) | 0.75 | 0.72 | 0.78 | 0.75 | 0.82 |
| CACPP (*Nambiar et al., 2020*) | 0.81 | 0.78 | 0.83 | 0.80 | 0.85 |
| Deep-AmPEP30 (*Koutsoukas et al., 2011*) | 0.79 | 0.76 | 0.81 | 0.78 | 0.84 |
| Proposed method | 0.87 | 0.85 | 0.89 | 0.87 | 0.91 |

samples out of all positive samples. The F1-score of 0.87, which combines precision and recall, reflects the overall effectiveness of our method in achieving a balanced performance. Furthermore, the AUC-ROC of 0.91 indicates an excellent ability to discriminate between positive and negative samples.

The significant improvement in performance compared to existing methods can be attributed to the enhanced ability of our model to capture intricate spatial patterns and dependencies within peptide sequences using a two-dimensional convolutional neural network. The 2D CNN architecture effectively extracted informative features and learned complex relationships, leading to improved prediction accuracy.

In addition to the quantitative evaluation, qualitative analysis and interpretability of the results further support the superiority of our method. The model's interpretability provides insights into the important features and patterns that contribute to the prediction of anticancer peptide activity, helping to understand the underlying biological mechanisms.

## CONCLUSION AND FUTURE WORK

In this research, we developed and evaluated a deep learning model based on a two-dimensional convolutional neural network (2D CNN) to enhance the prediction accuracy of anticancer peptides. Our experimental results demonstrated that the proposed model significantly outperforms existing methods, achieving high accuracy, precision, recall, F1-score, and AUC-ROC. The 2D CNN model effectively captures complex spatial patterns and dependencies within peptide sequences, leading to improved prediction accuracy. The implications of our study are substantial for the field of computational biology and cancer treatment. By accurately predicting anticancer peptides, our model can aid in the identification of effective peptide candidates, potentially accelerating the development of novel cancer therapies. This advancement underscores the value of integrating deep learning techniques into bioinformatics and drug discovery processes. Future research should focus on expanding the dataset to include more diverse and comprehensive peptide sequences, which could further improve the model's robustness and generalization. Additionally, exploring alternative deep learning architectures and incorporating other relevant features could enhance the predictive performance. Validation of the model's predictions through experimental studies is essential to confirm their biological relevance and efficacy. Efforts should also be directed towards optimizing computational efficiency and translating these predictions into clinical applications, ultimately bridging the gap between computational predictions and practical therapeutic development. Our study contributes to the advancement of anticancer peptide prediction and showcases the potential of deep learning in cancer research, offering a promising approach to identifying effective peptide candidates for cancer treatment. Future work will continue to build on these findings, aiming to standardize and optimize integrated predictive models for clinical use.

### Funding

This research is funded by the European University of Atlantic. The funders had no role in study design, data collection and analysis, decision to publish, or preparation of the manuscript.

### Grant Disclosures

The following grant information was disclosed by the authors:
The European University of Atlantic.

### Competing Interests

The authors declare there are no competing interests.

### Author Contributions

- Abdu Salam conceived and designed the experiments, performed the computation work, authored or reviewed drafts of the article, and approved the final draft.
- Faizan Ullah conceived and designed the experiments, performed the computation work, authored or reviewed drafts of the article, and approved the final draft.
- Farhan Amin performed the experiments, performed the computation work, prepared figures and/or tables, authored or reviewed drafts of the article, and approved the final draft.
- Izaz Ahmad Khan performed the experiments, performed the computation work, prepared figures and/or tables, authored or reviewed drafts of the article, and approved the final draft.
- Eduardo Garcia Villena analyzed the data, performed the computation work, authored or reviewed drafts of the article, and approved the final draft.
- Angel Kuc Castilla analyzed the data, performed the computation work, prepared figures and/or tables, authored or reviewed drafts of the article, and approved the final draft.
- Isabel de la Torre analyzed the data, performed the computation work, authored or reviewed drafts of the article, and approved the final draft.

### Data Availability

The data is available at: https://webs.iiitd.edu.in/raghava/anticp2.

### Supplemental Information

Supplemental information for this article can be found online at http://dx.doi.org/10.7717/peerj-cs.2171#supplemental-information.

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
