# Peer review of "Efficient prediction of anticancer peptides through deep learning"

_PeerJ Computer Science, doi:10.7717/peerj-cs.2171_

## Round 0.1 · original submission · Major Revisions

Please see the reviewers' detailed comments. The reviews of the manuscript highlight several significant areas for improvement. Firstly, the abstract has a consistent call for clarity and conciseness, emphasizing the need to focus on essential information such as the study's objectives, methods, and implications. Additionally, reviewers suggest including a summary of the study's implications and achievements in the conclusion section and future work. Concerns about paper organization and structure are raised, with recommendations to revise the presentation for better understanding. The need for more up-to-date references and clearer descriptions of experimental design and comparison methods is emphasized, along with the importance of discussing potential limitations. Finally, language issues and proofreading are mentioned, suggesting a thorough review to address these concerns. Addressing these points enhances the manuscript's overall quality and impact.

Reviewer 1 ·

Basic reporting

no comment

Experimental design

no comment'

Validity of the findings

This paper proposes a model for the discovery of anticancer peptides using deep learning. The topic is very interesting and will interest readers. However, there are various serious concerns related to this paper. Based on these concerns, I suggest a major revision for this paper. Please improve your paper; I want to see a clear improvement in the revised version. My comments and suggestions are given below.

Comments:
• The problem statement should be briefly described in the Abstract section. How it works, and the implications of this research should be mentioned in the abstract section.
• I have carefully checked the introduction section. The state of the art is not mentioned, and thus, the authors discuss the state of the art and its drawbacks.
• This research's problem statement and benefits should be highlighted in the introduction section.
• The contribution should be more polished and part of the introduction section.
• The research gap should be indicated. The article's aim should also be provided in the Introduction (not just in the Abstract).
• Section 3 is very important. I found that it comprises subsection 3.5. In my opinion, evaluation metrics should be shifted to section 4.
• The simulation environment and the parameters should be mentioned in the section 4.
• In the results, Figure 3 shows the “Performance Metrics of the proposed 2D CNN Architecture.” The title is not suitable. It should be modified with a suitable title. What the y-axis means should also be mentioned in the text.
• The conclusion section should be more précised.
• The paper does not discuss the impact of different hyperparameters on the model's performance.
• The authors should include a discussion on their proposed method's potential limitations and challenges.
• English is acceptable.

Additional comments

N/A

Cite this review as

Reviewer 2 ·

Basic reporting

In this paper, the author proposes a model using a two-dimensional CNN to predict anticancer peptides. They used deep learning for the proposed model. I agree that the discovery of anticancer peptides is a challenge. Overall, the paper is interesting and well-written. The topic is largely studied in the literature, but it is quite important, so I think such a manuscript can be of interest to the readers of this journal.
Please revise your paper according to the below suggestions.


1) The abstract and the conclusion section should be more polished. The conclusion section should include future work, and the abstract should briefly describe the research results.
2) The paper organization is missing at the end of the introduction section. Please mention this at the end of the introduction section.
3) I recommend that the authors complete a proofread of the text to remove problems with language and citations. The study and inclusion of the article ‘A Hybrid Deep Learning-Based Approach for Brain Tumor Classification’ will improve the content of the study.

Experimental design

1) The authors should properly describe the performance metrics based on the results.
2) The authors compared the proposed algorithm with other methods, but these methods have not been described, and no references are provided.
3) The authors do not describe the configuration of each compared method. The performance metrics used to evaluate the model's results should be clearly described. This will provide readers with a better understanding of how the results were measured and ensure the study's reproducibility.

Validity of the findings

1) While the authors compared the proposed algorithm with other methods, these methods lack proper descriptions and references. Including this information will strengthen the validity of the comparisons and provide a clearer context for the improvements claimed.
2) The paper does not detail the configuration settings for each compared method. This information is crucial for understanding the baseline conditions and ensuring that the comparisons are fair and replicable.

Cite this review as

Reviewer 3 ·

Basic reporting

no comment

Experimental design

no comment

Validity of the findings

no comment

Additional comments

I have gone through this study and found that it needs serious attention. Thus, it should be improved in terms of the following points.

1- There is unnecessary information in the abstract, thus, it should be polished. We expect a scientific manuscript presenting the study's results, not a report summarizing a study. I think you should see the following information in a few sentences in the abstract.
1.1. What was done in this study?
1.2. Why was this study done?
1.3. What are, the implications of this study?
1.4 The motivation and the contribution should be mentioned in the introduction section
The abstract section should answer the above questions and should be more polished.

2- A summary is required in the conclusion section. The implications of your study and the success achieved should be summarized in just a few sentences.
3- The paper presentation, organization, and structure of the article should be revised to make the subject more understandable.
4- Most references on this subject should be added. The number of references is not small. However, in this type of study, I think there should be much more up-to-date references.
5- I think that the Result, Discussion, and Conclusion sections should exist separately for such an article.
6- Authors are advised to add the latest citations, please read and cite.
7- Overall, there are still some minor parts that the authors did not explain clearly. There are typos and grammar mistakes and as a result, I am going to suggest a Major revision of the paper in its present form.

Cite this review as

---

## Round 0.2 · accepted · Accept

All reviewers confirm that the authors have addressed all of their comments.

Reviewer 1 ·

Basic reporting

No Comment

Experimental design

No Comment

Validity of the findings

No Comment

Additional comments

No Comment

Cite this review as

Reviewer 2 ·

Basic reporting

As a reviewer, I have no further comments and it seems that the article has been revised sufficiently for further processing towards acceptance as per the journal's policy. Thanks

Experimental design

The article carries results as needed for the journal's publication.

Validity of the findings

All data has been provided and conclusions are well stated. Thanks

Cite this review as

Reviewer 3 ·

Basic reporting

no comment

Experimental design

no comment

Validity of the findings

no comment

Additional comments

The authors have addressed my comments.

Cite this review as